

# Characteristics of soil profile $CO_2$ concentrations in karst areas and its significance for global carbon cycles and climate change

Qiao Chen*

Shandong Provincial Key Laboratory of Depositional Mineralization & Sedimentary Minerals, College of Earth Science &
Engineering, Shandong University of Science and Technology, Qingdao
Correspondence: Chen Qiao (qchen5581@163.com)

**Abstract:** $CO_2$ concentrations of 21 soil profiles were measured in Zhaotong City, Yunnan Province. The varying characteristics of soil profile $CO_2$ concentration are distinguishable between carbonate and non-carbonate areas. In non-carbonate areas, soil profile $CO_2$ concentrations increase and show significant positive correlations with soil depth. In carbonate areas, however, deep soil $CO_2$ concentrations decrease and have no significant correlations with soil depth. Soil organic carbon is negatively correlated with soil $CO_2$ concentrations in non-carbonate areas. In carbonate areas, such relationships are not clear. It means the special geological process in carbonate areas- carbonate corrosion- absorbs part of the deep soil profile $CO_2$. Isotope and soil pH data also support such process.

Mathematical model simulating soil profile $CO_2$ concentration was proposed. In non-carbonate areas, the measured and the simulated values are almost equal, while the measured $CO_2$ concentrations of deep soils are less than the simulated in carbonate areas. Such results also indicate the occurrence of carbonate corrosion and the consuming of deep soil $CO_2$ in carbonate areas. The decreased $CO_2$ concentration was roughly evaluated based on stratigraphic unit and farming activities. Soil pH and the purity of $CaCO_3$ in carbonate bedrock deeply affect the corrosion. The corrosion in carbonate areas decreases deep soil $CO_2$ greatly (accounting for 10-70%, with average of 36%), and naturally affects the soil $CO_2$ released into the atmosphere. Knowledge of this process is important for karst carbon cycles and global climate changes, and it may be a potential part of the "missing sink".

**Key words**: soil $CO_2$; carbonate corrosion; global carbon cycles; karst areas

## 1 Introduction

In recent years, there has been increasing world-wide concern about carbon exchange among the atmosphere, the ocean and terrestrial ecosystems. Specifically, there have been ongoing questions regarding the problem of carbon flux, or carbon source versus carbon sink. The missing carbon sink has puzzled scientists since Callendar (1938) presented the imbalance of absorbed and released $CO_2$. The missing sink reaches as much as 2-3 Pg, accounting for as much as 10-20% of total carbon (Jeremy et al. 2018). There are differing viewpoints regarding the spatial distribution and absorption strength of the missing sink in terrestrial ecosystems (Jeremy et al. 2018). The carbon cycle in karst areas has attracted great interest due to the absorbed and released $CO_2$ via carbonate corrosion and its share in regulating atmospheric $CO_2$. Therefore, some workers have looked for the "missing sink" within the absorbed and released carbon in karst systems, and the estimated values reach a dominating part (almost 1/3) of the missing sink (Jiang and Yuan, 1999).

Soil carbon, with storage of 1300-2000 Pg C, and as much as 2-3 times of vegetation storage, plays an important role in maintaining carbon balance (Fearnside, 2018), so that a slight change imposes a great effect on the atmospheric $CO_2$ concentration. Several factors affecting soil $CO_2$ concentration, such as environmental factors (soil temperature, moisture and water

---

*Corresponding author*. Qiao Chen, qchen5581@163.com



48 content etc) and human activities, have been widely discussed (Bajracharya et al., 2000; Dai et al.
49 2004; Jeremy et al. 2018; Fearnside, 2018). In karst areas, however, the important geological
50 process, carbonate corrosion, has been largely ignored in discussions of soil $CO_2$ levels, and
51 there is no detail documents detailing the soil $CO_2$ concentration and its relationship with
52 global climate change in karst areas. Several problems puzzle us: Is there any difference
53 between soil profile $CO_2$ concentrations in carbonate areas compared to that in non-carbonate
54 areas? If so, is the difference caused by carbonate corrosion? By how much is it affected?
55 Moreover, studies have revealed that there is $CO_2$ unbalance between carbon released into
56 atmosphere and that produced by organic matter in carbonate areas (Jiang and Yuan et al. 1999;
57 Jeremy et al. 2018), but there is no reasonable explanation. Lack of research work on these
58 questions restricts our understanding about soil $CO_2$ transfer, limits further study of the
59 mechanisms, and impedes learning of its significance for the carbon cycle.

60 In order to understand the varying characteristics of soil $CO_2$ concentration in karst areas
61 and its potential effect on global carbon cycles, soil profile $CO_2$ was measured, and samples of
62 soils and rocks were gathered in the typical karst area of Zhaotong city, Yunan Province, China.
63 The objectives of this paper are to: (1) analyze comparatively the varying characteristics of the
64 soil profile $CO_2$ concentration in carbonate and non-carbonate areas; (2) discuss the relationship
65 between soil $CO_2$ concentration and other parameters, and clarify the effect of carbonate
66 corrosion on soil $CO_2$; (3) develop a mathematical model of soil $CO_2$ transfer and
67 quantitatively evaluate the effect scale of carbonate corrosion on soil $CO_2$ concentration, and
68 discuss its significance for global carbon cycle and climate change.

## 2 Study area and methods
69

### 2.1 Study area
70

71 The study area, Zhenxiong County and Weixin County in Zhaotong City, north of the Yunnan
72 Province, China, was selected. The area contains high mountains and steep gorges. Many of the
73 mountain peaks tower above 2000 m, and there are many different natural watersheds. The
74 area is sub-tropical and humid. It has a plateau-climate with an average annual temperature of
75 11.7 °C and an average precipitation of 1200 mm. Monthly precipitation is above 100 mm, and
76 vertical climate belts with four seasons are clearly demarcated. The soil types include mainly
77 yellow, dingy and brown earth, with a wide thickness range (from a few up to 70-80 cm). The
78 flora is dominated by grass, shrubs, and partly by secondary forest.

79 The bedrock is composed predominantly of Mesozoic limestone and dolomite, with flysch
80 and associated sedimentary rocks. The widely exposed strata include mainly Ordovician,
81 Permian, Triassic, Jurassic and Quaternary units, among which only Ordovician and Permian
82 strata appear together. Devonian strata are not present, and Precambrian, Cambrian and Silurian
83 strata occur in limited outcrop, or as inclusions among other strata. Ordovician, Permian and
84 Triassic rocks are mainly marine carbonate deposits, and Jurassic and Quaternary units are
85 mainly composed of terrestrial clastic deposits.

### 2.2 Sampling and analyzing methods
86

87 In order to comprehensively reveal characteristics of soil $CO_2$ concentration in a karst area,
88 sample sites were selected in such a way as to cover different stratigraphic units and different
89 types of vegetation. Meanwhile, profiles in carbonate and non-carbonate areas were both
90 measured. Totally, $CO_2$ concentration of 21 soil profiles and organic carbon of 12 soil profiles
91 were analyzed. The profile sites are shown in Fig. 1, and among these, profiles in carbonate
92 areas include the Lower Ordovician Meitan Formation ($O_1$), the Middle and Upper
93 Ordovician Baota Formation ($O_{2-3}$), the Lower Permian Xixia and Maokou Formations



(P₁m(q)), the Upper Permian Changxing Formation (P₂c), and the Middle Triassic Guanling Formation (T₂g). Sites in non-carbonate areas include Middle Permian basalt (P₂β), shale in the Upper Permian Longtan Formation (P₂l), mudstone in the Lower Triassic Feixianguan Formation (T₁f), and siltstone intercalated with shale in the Upper Tirassic Xujiahe Formation (T₃x).

CO₂ concentration within the soil pores was measured every 10 cm from the surface down to the rock-soil interface using a GASTEC 801 instrument and 2LL or 2L CO₂ Detector Tube (GASTEC Co., Japan). The profile soil samples were of one-to-one correspondence with the gas samples and also taken every 10 cm.

The starting samples were air-dried naturally, and then pulverized (particle diameter <150 μm). Soil organic carbon was determined using the potassium dichromate volumetric method. Soil pH was measured in distilled water at a solid/ solution ratio of 1/5, with the instrument model PHS-2. Water contents of soils were synchronously measured by a cutting ring. CaO and MgO contents of rocks were determined by Inductively Coupled Plasma-Atomic Emission Spectrometry (ICP-AES) with a Charge Injection Detector (CID), model TJA IRIS/AP. The standard materials (GBW07401, GBW07408) were used for quality control, with relative deviation less than 5%.

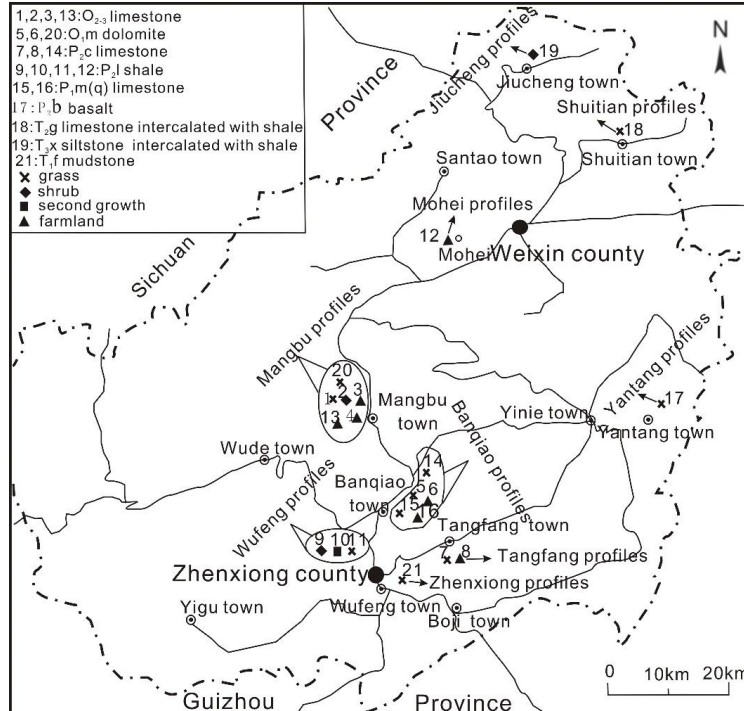

FIG.1. Sites of measuring soil CO₂ and gathering organic carbon samples (1-Mangbu O₂₋₃ grass, 2-Mangbu O₂₋₃ shrub, 3-Mangbu O₂₋₃ farmland, 4-Mangbu O₂₋₃ farmland, 5-Banqiao O₁m grass, 6-Mangbu O₁m farmland, 7-Tangfang P₂c grass, 8-Tangfang P₂c farmland, 9-Wufeng P₂l shrub, 10-Wufeng P₂l second growth, 11-Wufeng P₂l grass, 12-Mohei P₂l farmland, 13-Mangbu O₂₋₃ farmland, 14-Banqiao P₂c grass, 15-Banqiao P₁m(q) grass, 16-Banqiao P₁m(q) shrub, 17-Tangfang P₂β grass, 18-Shuitian T₂g shrub, 19-Jiucheng T₃x shrub, 20-Mangbu O₁m grass, 21-Zhenxiong T₁f grass).

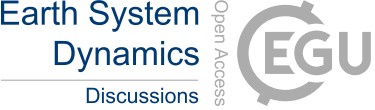

## 3 Results

### 3.1 Varying CO₂ concentration characteristics of soil profiles

Fig. 2 shows soil profile $CO_2$ concentrations varying with soil depths in 7 non-carbonate areas. The data show a distinct tendency of increasing $CO_2$ concentration with soil depth, with $R^2$=0.8-0.92 (Table 1).The reasons may be the higher soil bulk density, more condensed soil pores, and difficulty of $CO_2$ diffusion in the deeper soil. In fact, soil profile $CO_2$ has been widely reported to be correlated with soil depth by previous researches (Rustad et al. 2000; Dai et al. 2004; Malak et al. 2018) , and even the following linear equation have been developed (James and George, 1991): *Mean CO₂=0.035+0.0015(Depth) (R²=0.99, P<0.0005)*. Our observations in non-carbonate areas are concordant with these reports and support soil profile $CO_2$ increases with soil depth in non-carbonate areas.

Table 1. Regression analysis of soil $CO_2$ concentration and profile depth in non-carbonate areas.

| Profiles | Regression equation | $R^2$ | P |
|---|---|---|---|
| Wufeng P₂l shrub (9) | y = 0.0077x + 0.7692 | 0.92 | 0.179 |
| Wufeng P₂l second growth (10) | y = 0.0099x - 10.595 | 0.80 | 0.016* |
| Wufeng P₂l grass (11) | y = 0.0015x + 11.527 | 0.80 | 0.042* |
| Mohei P₂l farmland (12) | y = 0.0031x + 12.239 | 0.80 | 0.039* |
| Yantang P₂ß grass (17) | y = 0.0415x - 19.114 | 0.85 | 0.077 |
| Zengxiong T₁f grass (21) | y = 0.15x - 70 | 0.9 | 0.051 |
| Jiucheng T₃x shrub (19) | y = 0.0086x + 5.6875 | 0.81 | 0.101 |

Note: ( ) means profile No., *means significantly congressed at 0.005 level.

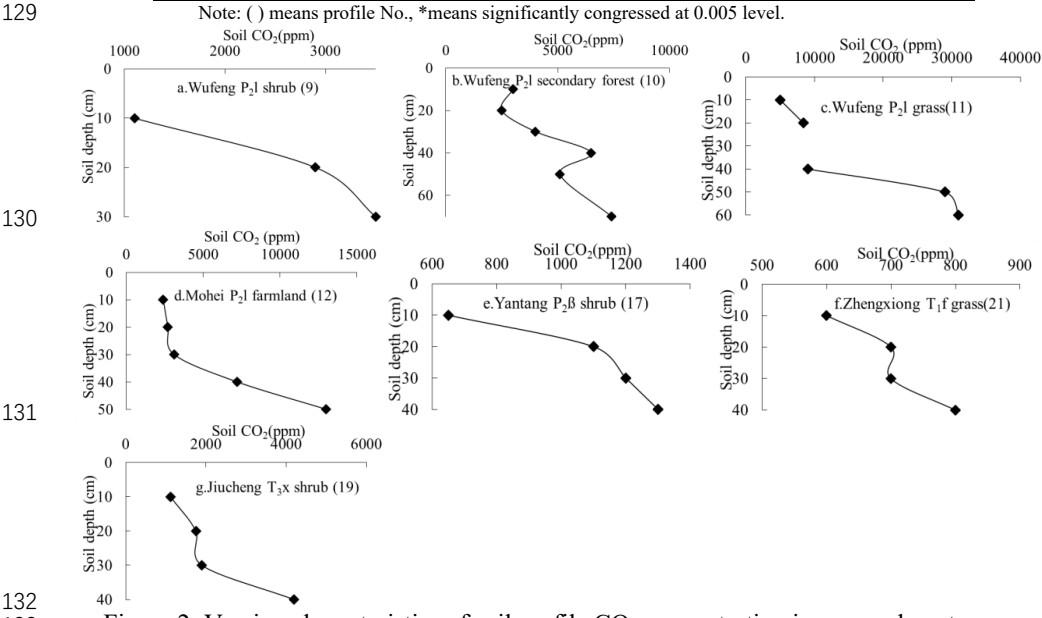

Figure 2. Varying characteristics of soil profile CO₂ concentration in non-carbonate areas
(profile no. in brackets).



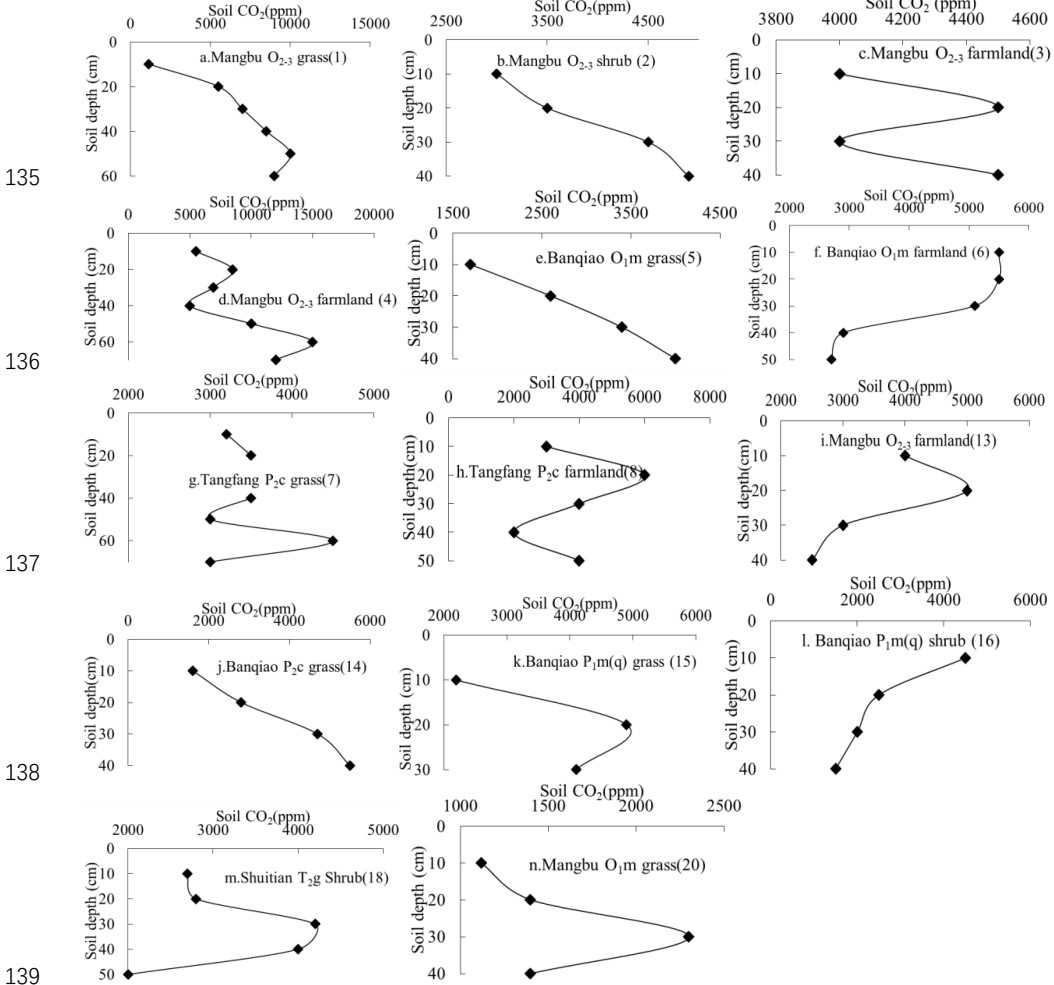

Figure 3. Varying characteristics of soil profile $CO_2$ concentration in carbonate areas (profile no. in brackets).

14 Soil profile $CO_2$ concentration with soil depth in carbonate areas was gained (Fig. 3). The results show a complex and inverse relationship between soil $CO_2$ and soil depth in carbonate areas. Most Soil profile $CO_2$ increases with soil depth in the upper sections, such as Mangbu $O_{2-3}$ grassy profile (Fig. 3a), Mangbu $O_{2-3}$ shrub profile (Fig. 3b), Mangbu $O_{2-3}$ farmland profile (Fig. 3d), Banqiao $O_1m$ grassy profile (Fig. 3.e), and Banqiao $P_2c$ grassy profile (Fig. 3j). $CO_2$ concentrations decrease with soil depth when they increase from surface to a certain depth in Mangbu $O_{2-3}$ farmland profile (Fig. 3i), Banqiao $P_1m(q)$ grassy profile (Fig. 3k), Gaotian $T_2g$ grassy profile (Fig. 3m) and Mangbu $O_1m$ grassy profile (Fig 3n). Those of Banqiao $O_1m$ farmland profile (Fig. 3f) and Banqiao $P_1m(q)$ shrub profile (Fig. 3l) even decrease all along with soil depth, and two farmland profiles of Mangbu $O_{2-3}$ (Fig. 3c) and Tangfang $P_2c$ (Fig. 3h) fluctuate, and have no regularity due to the effect of human farming activities. Generally, Except Mangbu $O_{2-3}$ farmland profile (Fig. 3.c) and Tangfang $P_2c$ farmland profile (Fig. 3.h), which are disturbed by farming, $CO_2$ concentrations of other profiles in carbonate areas all decrease with soil depth at the rock-soil



interface (Fig. 3.b,e,j). Moreover, there is no correlation of soil $CO_2$ concentration with soil depth,
because sequestration of deep soil $CO_2$ concentration occurs in carbonate areas. Why does the
sequestration only take place in carbonate areas, but not in non-carbonate ones? Naturally the
particular carbonate process-carbonate corrosion-is considered. That is, part of deep soil $CO_2$ is
consumed and $CO_2$ sequestration occurs, and there is no linear relationship between $CO_2$
concentration and soil depths in carbonate areas. In fact, Buyannovsky and Wagner (1983),
Solomon and Cerling (1987), and Xu and He (1996) all reported that soil $CO_2$ concentration
reaches a peak at a certain depth, and then decreases with soil depth in carbonate areas. $CO_2$
concentration in Banqiao $O_1m$ farmland profile (Fig. 3.f) and Banqiao $P_1m(q)$ shrub profile (Fig.3.l)
continues to decrease with depth through the integral profile, and they also had the highest
concentration at the 10cm layer. Instances of $CO_2$ concentration in surface layers higher than those
in bottom layers are scarcely documented in carbonate areas.
**3.2 Relationship between soil profile $CO_2$ concentration and soil organic carbon**
Soil organic carbon (SOC) was analyzed in a part of the profiles, corresponding with $CO_2$
concentration. Results are given in Fig. 4 a-h, indicating profiles in carbonate areas, whereas
Fig. 4 i-l indicate those in non-carbonate (shale) areas.
Correlation analysis of soil profile $CO_2$ concentration and SOC in shale areas is listed in
Table 2. It shows a negative correlation, with high regression coefficients ($R^2$= 0.67-0.85). An
exception of 0.29 occurs in Wufeng $P_2l$ secondary forest, which possibly is caused by stronger
root respiration and a higher ratio of $CO_2$ generated by the roots. Therefore, SOC is directly
affected by the release of soil $CO_2$, and the key problem for soil carbon storage is to slow down
the renewing of soil organic matter (Chen et al. 2002). However, the soil profile $CO_2$
concentration don't show significant regression with SOC, which meanswhich means that soil
$CO_2$ concentration is not only related to SOC, but also to soil respiration and microbe activities.
Correlation analysis of soil $CO_2$ and organic carbon in carbonate areas is shown in Table 3, and
the regression coefficients are irregular, and even those of Banqiao $O_1m$ farmland profile and
Banqiao $P_1m(q)$ shrub profile are positive. Previous studies in Shilin, Lunan City and in
Guizhou Plateau also showd no correlation between $CO_2$ concentration and SOC (Liang et al.
2003).

Table 2. Correlation analysis of soil $CO_2$ and soil organic carbon in shale areas of karst.

| Profiles | Regression equation | $R^2$ | P |
|---|---|---|---|
| Wufeng $P_2l$ shrub (9) | y = -618.67x + 4199.6 | 0.67 | 0.387 |
| Wufeng $P_2l$ second growth (10) | y = -766.39x + 7548.9 | 0.29 | 0.239 |
| Wufeng $P_2l$ grass (11) | y = -13093x + 69890 | 0.74 | 0.351 |
| Mohei $P_2l$ farmland (12) | y = -8646.2x + 49490 | 0.85 | 0.077 |

Table 3. Correlation analysis of soil $CO_2$ and soil organic carbon in carbonate areas of karst.

| Profiles | Regression equation | $R^2$ |
|---|---|---|
| Mangbu $O_{2-3}$ grass (1) | y = -4673.8x + 15214 | 0.35 |
| Mangbu $O_{2-3}$ shrub (2) | y = -1054.5x + 5273.4 | 0.46 |
| Mangbu $O_{2-3}$ farmland (3) | y = -61.209x + 4305.9 | 0.005 |
| Mangbu $O_{2-3}$ farmland (4) | y = -3569.5x + 10875 | 0.25 |
| Banqiao $O_1m$ grass (5) | y = -1172.2x + 8636.5 | 0.68 |
| Banqiao $O_1m$ farmland (6) | y = 5560.6x - 639.97 | 0.84 |
| Tangfang $P_2c$ grass( 7) | y = -134.06x + 3594.1 | 0.33 |
| Tangfang $P_2c$ farmland (8) | y = 4477.3x - 2714.1 | 0.44 |

What is the reason of poor relationship between soil $CO_2$ and SOC in carbonate areas? The
possible answer may be carbonate corrosion. By means of corrosion, deep soil $CO_2$ is partly



consumed and its level decreases. Consequently, the relationship becomes poor. In addition,
varying characteristics of SOC cannot explain well the decrease of deep soil $CO_2$ levels in
carbonate areas.



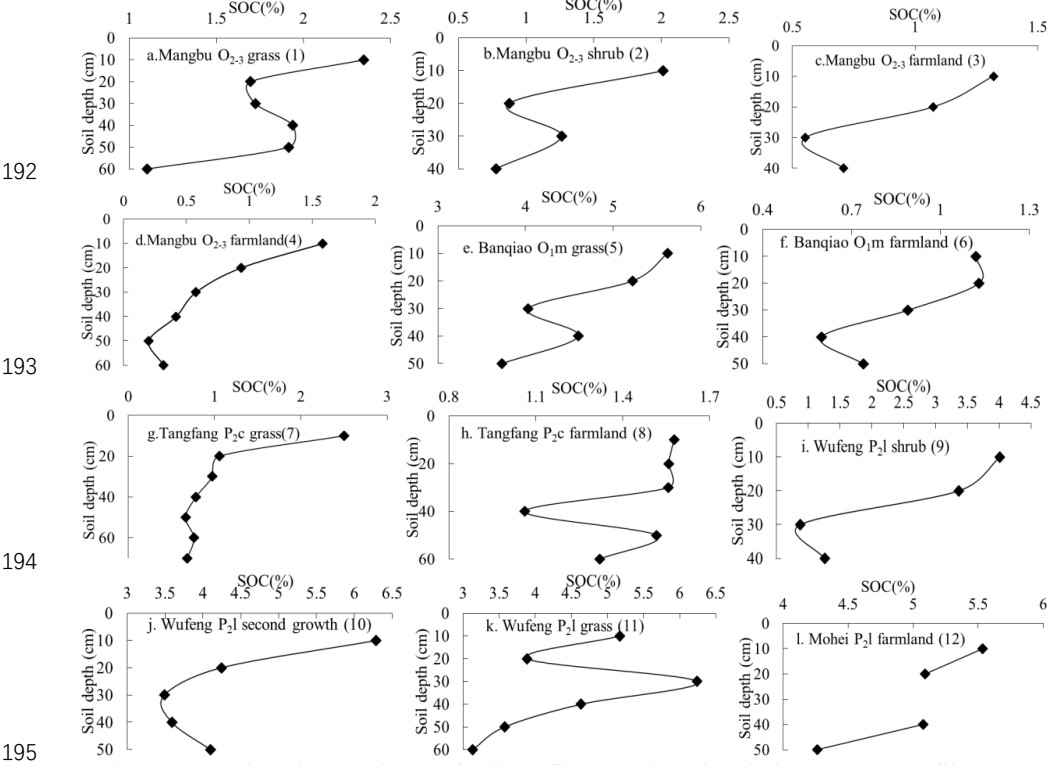

Figure 4. Varying characteristics of soil profile organic carbon in karst areas (profile no. in
brackets)
**3.3 Varying characteristics of profile soil pH**
Soil pH curves varying with soil depth are drawn in Fig. 5a-h, indicating carbonate profiles,
and 6i-l indicating non-carbonate (shale) profiles. In non-carbonate areas, there is a complex
relationship between pH and depths, but pH increases obviously at the rock-soil interface,
whereas pH non-significantly varied with soil $CO_2$ and SOC. Conversely, in carbonate areas,
pH generally increases with soil depth in the surface layer except in the Banqiao $O_1m$ farmland
profile. Moreover, from Figs. 3 and 5 it is evident that soil $CO_2$ concentration decreases where
soil pH decreases too, and even $CO_2$ level in the Banqiao $O_1m$ farmland profile decreases from
the surface to the bottom with soil pH through the entire profile. These observations imply that
the decrease of deep soil $CO_2$ concentration in carbonate areas is related closely to soil pH.
Chemically, with soil water and soil $CO_2$ added together, carbonate corrosion can be
represented by the following equation:
$$CaCO_3 + CO_2 + H_2O <=> Ca^{2+} + 2\ HCO_3^-$$
$$HCO_3^- <=> H^+ + CO_3^{2-}$$
By means of this reaction, deep soil $CO_2$ is consumed by the corrosion of the underlying
carbonate rock, and pH decreases synchronously. This reaction cannot take place in soil over





areas with non-carbonate bedrock, so here the deep soil $CO_2$ concentration does not decrease,
but increases.

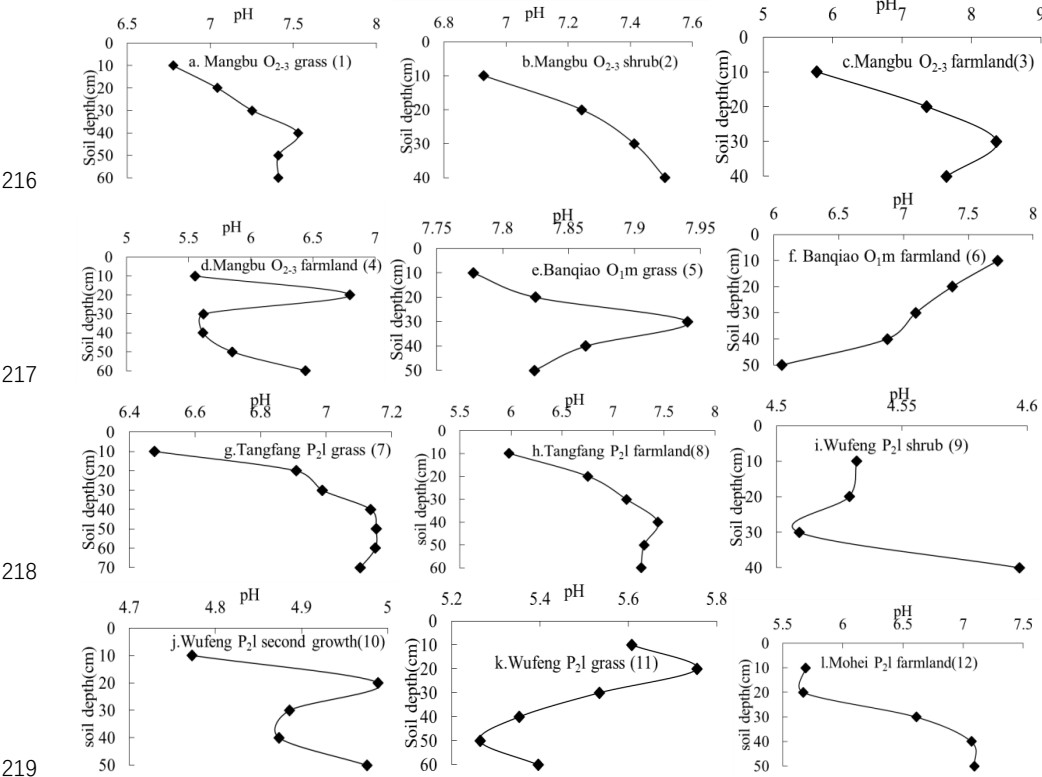

Figure 5. Soil pH of different profiles in karst area (a-h indicate those in carbonate areas, and
h-l indicate those in shale areas, profile no. in brackets).

## 3.4 Carbonate corrosion and the global carbon cycle

Many studies have observed that soil $CO_2$ concentration in carbonate areas decreases with depth
when it reaches a maximum at a certain soil depth in carbonate areas (Buyannovsky and
Wagner, 1983; Li et al. 1995; Xu and He, 1996; Liang et al. 2003). There has, however, been
no reasonable explanation for the observations. Li et al (1995) attributed it to less roots, and,
therefore, less root respiration in the deep soil, but there are no scientifically observed data to
support this idea, and it remains only a hypothesis. No decrease in soil $CO_2$ in non-carbonate
areas is found, and, furthermore, the depths with decreasing $CO_2$ concentrations were
distinguishable in different profiles, even at only 20-30 cm depths. The decreased $CO_2$
concentration could be attributed to decreased microbe numbers or root respiration at such depths.
By comparative analysis of soil $CO_2$ concentration in areas of carbonate and non-carbonate
bedrock, it should be suggested that the explanation is due to the special geological process of
carbonate corrosion.
Soil $CO_2$ and SOC in non-carbonate areas have a good negative correlation, with correlation
coefficients $R^2$= 0.67-0.85, although significance is not clear because soil $CO_2$ is determined by
not only organic matter but also by other factors, such as root respiration and microbe activities.
By contrast, such correlation in carbonate areas is poor, which was concluded also by Li et al



(1995) and Liang et al. (2003) from experiments in carbonate areas. Soil $CO_2$ of carbonate
areas, in every depth at different sites, is negatively correlated with SOC, and relationship
became worst with increasing soil depth. This observation means that SOC content cannot
explain well the decreased $CO_2$ concentration of deep soil in carbonate areas, but rather may be
related to carbonate corrosion. Soil pH in carbonate areas always decreases with soil $CO_2$, and
this may imply that $H^+$ generated by carbonate corrosion mixes into the deep soil increasing
soil acidity.
Previous work has determined the imbalance between soil $CO_2$ produced and released in
carbonate areas. Pan et al (2000) observed and simulated field data in Yaji, Guangxi Province,
concluding that $CO_2$ produced by decomposition of organic matter is more than that released
into the air. This confirms that the rock and the soil have an obviously "absorbing effect" for
$CO_2$. The data account for an absorbing coefficient of 22-130 $g/m^2 \cdot a$.
Isotopes can effectively trace the carbon source of soil $CO_2$. Fig. 6 reflects the $\delta^{13}C$ value
of soil $CO_2$ and SOC overlying different bedrock according to data from Li et al. (1995). It
shows that in deep soil, $CO_2$ has a higher $\delta^{13}C$ value than the SOC in limestone and dolomite
areas, whereas the isotope ratios are more equivalent in clay stone areas. Such an observation
may support the conclusion that that deep soil $CO_2$ in clay stone areas is mainly or completely
from soil organic matter, and that in limestone and dolomite areas there must be an additional
carbon source whose $\delta^{13}C$ should be more than -14‰ . $CaCO_3$ in carbonate has $\delta^{13}C$ values of -
3‰~+1‰. It must, therefore, possibly be recognized that carbon in $CaCO_3$ of carbonate bedrock
mixes into soil $CO_2$, since the corrosion reaction is reversible.

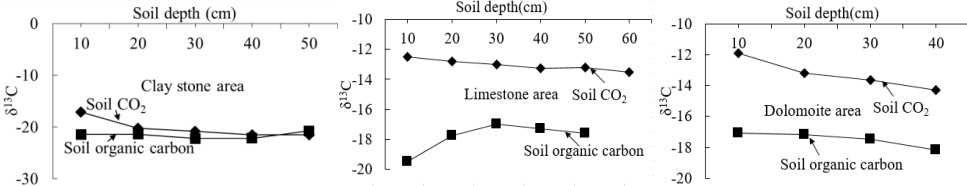

Figure 6. Varying $\delta^{13}C$ of soil $CO_2$ and soil organic carbon with soil depth overlying different
bedrocks (data is after Li et al (1995)).
It has been examined that the karst carbon cycle is an important trace for the global carbon
cycle and that further study is important to the hunt for "missing sink" (Jiang and Yuan,1999).
From what is presented above, with focus on the process of carbonate corrosion and comparison
of different parameters in carbonate and non-carbonate areas, it is logical to conclude that
carbonate corrosion causes the decreased $CO_2$ concentration at the rock-soil interface in
carbonate areas. As a result, the decreased $CO_2$ level caused by corrosion will, of course,
impose effects on atmospheric $CO_2$ and the karst carbon cycle. This is significantly for the
potential fixation of carbon, the study of global carbon cycle balance, and the hunt for the
"missing sink".

## 3.5 Mathematical model of soil profile $CO_2$ transfer

In this model, only the molecular diffusion of $CO_2$ is considered, neglecting other processes, such
as viscous flow and Knudsen diffusion in karst soil because of the weak air pressure gradient.
Moreover, density gradient was regarded as the dominant dynamic of $CO_2$ diffusion, and
temperature gradient was neglected because of its low contribution (0.2-0.4%) to $CO_2$ flow.
Therefore, the transport of soil $CO_2$ can be described by the following one-dimensional diffusion
equation according to Fick's second law and laws of conservation of mass (Zeng and Zheng,
2002), assuming horizontal homogeneity:



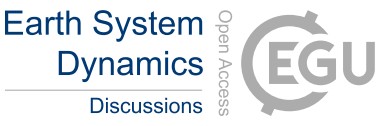

$$\frac{\partial(\theta_a C_a + \theta_w C_w)}{\partial t} = -\frac{\partial(J_{da} + J_{dw} + J_{ca} + J_{cw})}{\partial z} - Q \cdot C_w + S \qquad (1)$$
Here, $\theta_a$ is the air content, $\theta_w$ is the water content, $C_a$ is the gaseous $CO_2$ concentration, $J_{da}$ is the
gaseous $CO_2$ flow due to diffusion, $J_{dw}$ is the dissolution $CO_2$ flow due to diffusion, $J_{ca}$ is the
gaseous $CO_2$ flow due to convection, $J_{cw}$ is the dissolution $CO_2$ flow due to convection, S is the
carbon source, Q is the water absorbed by roots, t is the time, and z is the space coordinate.
Such equation can be gained according to Fick's first law:
$$J_{da} = -D_a \frac{\partial C_a}{\partial z} \qquad J_{dw} = -D_w \frac{\partial C_w}{\partial z} \qquad J_{ca} = q_a C_a \qquad J_{cw} = q_w C_a \qquad (2)$$
where $D_a$ is the gaseous $CO_2$ diffusion coefficient in soil substrate, $D_w$ is the dissolution $CO_2$
diffusion coefficient in soil substrate, $q_a$ is the soil air transference amount, and $q_w$ is the soil
water transference amount.
Equation (3) can be deduced from equations (1) and (2), if it is assumed that soil water is
stable and gaseous and dissolution $CO_2$ flows are not considered:
$$\theta_a \frac{\partial C_a}{\partial t} = D_a \frac{\partial^2 C_a}{\partial z^2} - \theta_w \frac{\partial C_w}{\partial t} - q_w \frac{\partial C_w}{\partial z} - Q \cdot C_w + S \qquad (3)$$
Previous studies were referenced when the parameters were determined, and all the
parameters should be gained in winter of the same working period:
$q_w = \tau \exp(-\frac{z}{\delta})$, presented by Yoyam et al.(1993), and
$Q = \dfrac{\tau \exp(-\frac{z}{\delta})}{\delta}$ by Warren and Michael (1984), and
in winter $q_w = 0$, $Q = 0$; $D_a = D_a^\circ (\frac{\theta_a}{\theta_w})(\theta_a)(\frac{T}{T_0})^{1.823}$, by Collin and Rasmuson (1988). Here, $D_a^0$ is
the $CO_2$ diffusion coefficient in air at the reference temperature $T^0$.
For the carbon source, the rate of $CO_2$ produced by root respiration and microbes can be
expressed as follows:
$S(z) = S_0 \exp(-z/z_s)$
where $S(z)$ is the soil profile $CO_2$ at depth of z, $S_0$ is the $CO_2$ concentration in the surface soil, z
is the soil depth, and $z_s$ is the depth gradient. It also considered the $CO_2$ produced by organic
matter expressed as follows:
$S_{OM} = -\dfrac{6 D_a \partial^2 C_a}{3.3 \partial z^2}$.
Then equation (4) is achieved:
$$\theta_a \frac{\partial C_a}{\partial t} = -0.82 D_a \frac{\partial^2 C_a}{\partial z^2} - \theta_w \frac{\partial C_w}{\partial t} + S_o \exp(-z/z_s) + a \qquad (4)$$
$\frac{\partial C_a}{\partial t}, \frac{\partial C_w}{\partial t}$ are stable, when being from the same time and soil profile.
Based on the studies above, the soil profile $CO_2$ concentration varying with soil depth can be
expressed by the following equation:
$C_a = A \exp(Bz) + Cz + D$   (A, B, C, D = uncertain)  (5)
According to Tailor formula:

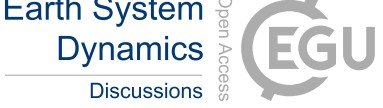



$\exp(x) = 1 + x + \frac{1}{2!}x^2 + \cdots\cdots + \frac{1}{n!}x^n + \cdots\cdots$ (6), and it can be roughly expressed like the following
equation when x<1:
$$\exp(x) = \begin{cases} 1+x & (x \ll 1) \\ 1+x+\frac{1}{2!}x^2 \end{cases} \qquad (7)$$
When equation (7) is applied to equation (5), equation (8) can be gained to express profile $CO_2$
concentration ($C_a$) varying with soil depth (z):
$$C_a = \begin{cases} a+bz \\ a+bz+cz^2 \end{cases} \qquad (8)$$
Here, a, b and c are uncertain parameters, which vary with $\theta_a$, $\theta_w$, $S_0$, T, and $D_a$ of different
profiles.
That means, it can be expressed as a linear or parabolic relationship of soil profile $CO_2$
concentration and soil depth. Actually, many observation and simulation also confirmed the
same results (James and George, 1991; Zeng and Zheng, 2002; Malak et al., 2018). Therefore,
it seems reasonable to express a linear or parabolic relationship of soil profile $CO_2$ concentration
and soil depth.

### 3.6 The rough evaluation of $CO_2$ decreased by corrosion

SPSS software was used to simulate the curve of measured soil $CO_2$ concentration and soil depth
in non-carbonate areas (Fig. 7 and Table 4), resulting in parabolas with multiple regression
coefficients $R^2$=0.8-1. Multiple regression coefficient of P2c secondary forest profile shows the
lowest level at 0.79, which may be due to the different root respiration and the absorbed water
at different depths. The simulation evidences that the model is reliable and can be used to
roughly reveal the laws of soil profile $CO_2$ concentration.
Table 4. Simulated equation of measured soil $CO_2$ concentration and soil depth in non-carbonate areas.

| Profiles | Equations | $R^2$ | P | Simulated depth | Simulated equation by exponents |
|---|---|---|---|---|---|
| $P_2l$ shrub (9) | y=-6x²+360x-1900 | 1 | - | 0-30 cm | y=702.44e^{0.0579x}(0.8681) |
| $P_2l$ second growth (10) | y=-0.1548x²+92.952x +1610 | 0.7924 | 0.0946 | 0-60 cm | y=2320.4e^{0.0175x}(0.7784) |
| $P_2l$ grass (11) | y=12.458x²-324.64x +7736.4 | 0.8673 | 0.1327 | 0-60 cm | y=3456.1e^{0.0363x}(0.8601) |
| $P_2c$ farmland (12) | y =10.5x²-373x +5320 | 0.9914 | 0.0086 | 0-50 cm | y=1221.3e^{0.0436x}(0.8877) |
| $P_2$ ß grass (17) | y=-0.875x² +64.25x+112.5 | 0.9752 | 0.1575 | 0-40 cm | y=597.91e^{0.0217x}(0.7989) |
| $T_1f$ shrub (21) | y=-4E-15x²+6x+550 | 0.9 | 0.3162 | 0-40 cm | y=561.25e^{0.0086x}(0.8977) |
| $T_3x$ shrub (19) | y=4.175x²-114.85x +1982.5 | 0.93 | 0.2519 | 0-40 cm | y=722.96e^{0.0405x}(0.9031) |

Note: regression coefficients $R^2$ of simulated exponent in brackets.
In carbonate areas, however, there is no linear or parabolic relationship between soil profile
$CO_2$ concentration and soil depth, and the measured values are inconsistent with the simulated
ones. Linear or parabolic relationship can be found in the surface soil. Since it is carbonate
corrosion that decreases the $CO_2$ concentration in the deep soil of carbonate areas, the $CO_2$
concentration in the surface layer can be used and to predict the $CO_2$ concentration of deep soil
based on the developed model. The predicting equation and results are listed in Fig. 8. It shows
that there is a strong difference between the measured and the predicted values, and that all the
predicted are greater than the measured ones in deep soil. It can also be deduced that deep soil
$CO_2$ is consumed by carbonate corrosion.
The method of subtraction of predicted and measured values can be used to evaluate the
decreased $CO_2$ concentration in carbonate areas caused by carbonate corrosion, and the results
are listed in Table 5. If synthesis factors, such as vegetation types and soil types, were

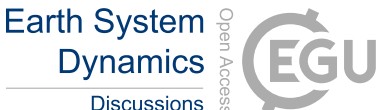

considered, the rough evaluation of the decreased $CO_2$ concentration of every stratigraphic unit
can be gained by taking the average (Fig. 9).
Table 5. The evaluated results of the decreased $CO_2$ concentration in carbonate areas caused
by carbonate corrosion.

| Profiles | $O_{2-3}$ grass (1) | $O_{2-3}$ shrub (2) | $O_{2-3}$ farmland (3) | $O_{2-3}$ farmland (4) | $O_1m$ grass (5) | $O_1m$ farmland (6) | $P_2c$ grass (7) | $P_2c$ farmland (8) | $O_{2-3}$ farmland (13) | $P_2c$ grass (14) | $P_1m(q)$ grass (15) | $P_1m(q)$ shrub (16) | $T_2g$ shrub (18) | $O_1m$ grass (20) |
|---|---|---|---|---|---|---|---|---|---|---|---|---|---|---|
| Decreased $CO_2$ concentration (ppm) | 2500 | 266.7 | 2000 | 1493.1 | - | 8800 | 1918.1 | 2600 | 7500 | 633.3 | 3500 | 10500 | 11800 | 2420 |
| Percentage of total deep soil $CO_2$ (%) | 21.7 | 5.2 | 19.0 | 6.2 | - | 48.9 | 39.0 | 14.4 | 57.7 | 10.3 | 46.0 | 63.6 | 66.3 | 63.4 |




Figure 7. The measured and the simulated $CO_2$ concentrations of soil profiles in non-



carbonate areas.








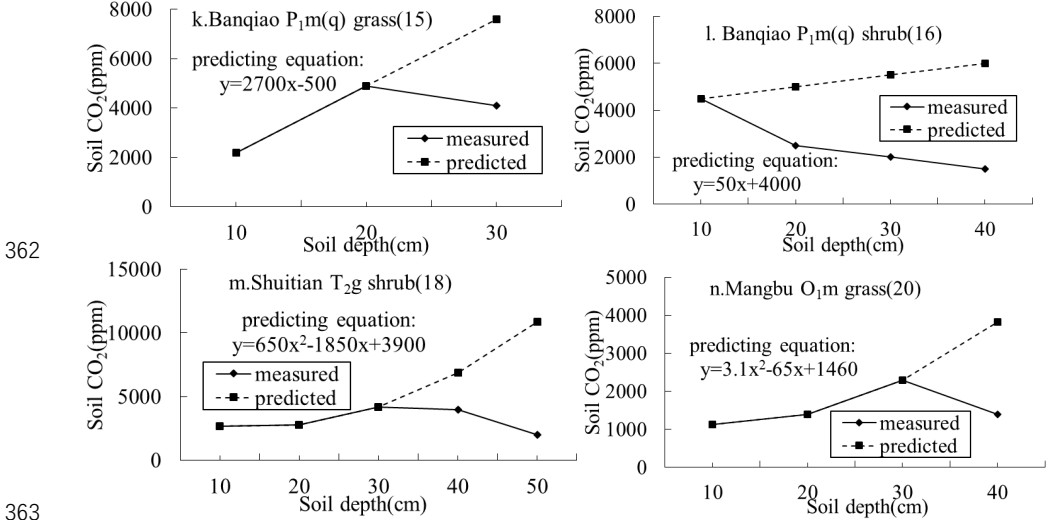


Figure 8. The measured and predicted soil profile $CO_2$ concentrations in carbonate areas.

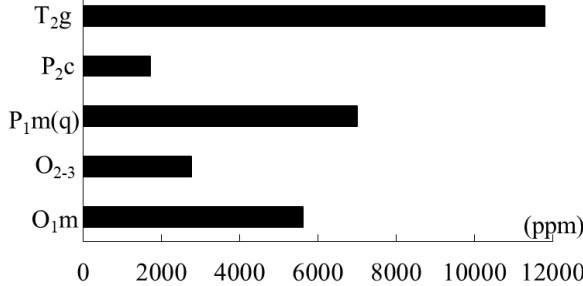

Figure 9. The evaluation of the decreased $CO_2$ concentration caused by carbonate corrosion
based on stratigraphic units.

**3.7 The main affecting factors of the decreased $CO_2$ concentration**
Fig. 9 shows great dissimilarity of the decreased $CO_2$ concentration with different stratigraphic
units in the following order: $T_2g>P_1m(q)>O_1m>O_{2-3}>P_2c$. Fig. 10 shows the calculated results
of the decreased $CO_2$ concentration, respectively, in farmland and natural soil (grass and shrub)
of the same stratigraphic unit. $CO_2$ concentration on $T_2g$ and $P_1m(q)$ farmland is lacking, but
the comparative analysis of $O_1m$ , $O_{2-3}$ and $P_2c$ can demonstrate that the decrease of $CO_2$ in
natural soil profiles is obviously less than that in farmland profiles. It is clear that corrosion was
strengthened by farming activities and more $CO_2$ was consumed in the deep soil, which may be
due to higher $CO_2$ levels and acidity caused by farming. Therefore, the decreased $CO_2$
concentrations of $T_2g$ and $P_1m(q)$ should be more than the calculated values, when farming
activities are considered. The decreased $CO_2$ concentration in different farmland profiles is
remarkably distinguishable at different sites, even on their same stratigraphic units (Table 5). It
seems that the degree of human activity and the quantities of imported or exported energy
determine the corrosion to some degree.
Several parameters, such as CaO and MgO contents of carbonate, water content and pH of
the overlying soil, were determined to address some natural factors affecting de-creased $CO_2$



concentration. The parameters are shown in Fig. 11. Deep soil-pH is negatively correlated with
decreased $CO_2$ concentration, and the stronger the soil acidity, the more the decreased $CO_2$
concentration. Water content of deep soil does not impose effort to corrosion. CaO content of
carbonate is positively correlated with the de-creased $CO_2$ concentration, and the more pure the
$CaCO_3$ in carbonate rock, the stronger is the corrosion. MgO content of carbonate is not
correlated with corrosion, which indicates that it is $CaCO_3$ corrosion and not that of $MgCO_3$
consuming soil $CO_2$. Simulation by SPSS software results in an equation (y=-3E -
$08x^2$+0.0002x+6.976) of decreased $CO_2$ concentration and soil pH with a multiple regression
coefficient $R^2$=0.9779, and a second equation (y=0.0012x +17.857) of decreased $CO_2$ level and
CaO content of carbonate with a multiple regression coefficient $R^2$ = 0.4191 (Fig. 12). A field
experiment of carbonate corrosion in the southern part of Guizhou (Nie et al.1984), a laboratory
simulation using citric acid to corrode limestone (Cao et al., 2001), and an experimental study on
the stability of $CaCO_3$ and $MgCO_3$ under acid rain conditions (Teir et al. 2006) led to the
conclusion that corrosion is related closely with soil acidity and carbonate purity. The calculated
results can support the same conclusion and accord well with their studies, and can also easily be
confident.

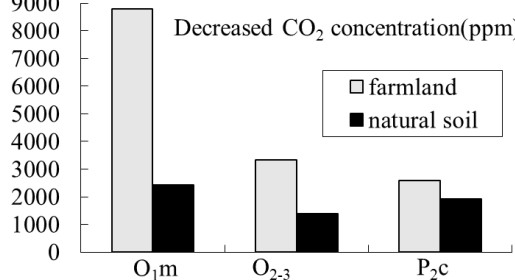

Figure 10. The decreased $CO_2$ concentration in farmland and natural soil of the same
stratigraphic unit.

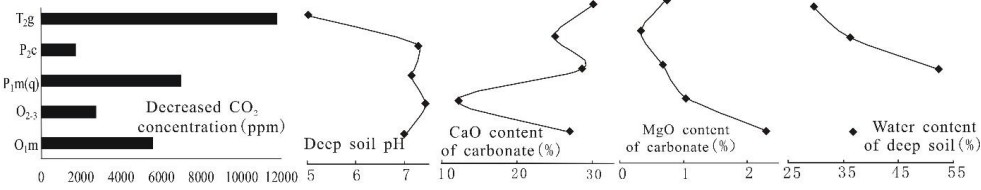

Figure 11. Relationship of the decreased $CO_2$ concentration and deep soil pH, water content,
CaO and MgO contents of carbonate.

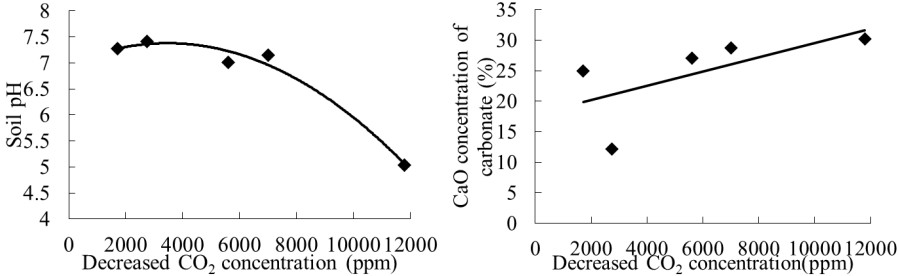

FIG. 12. Correlation analysis of soil pH, CaO of carbonate and decreased $CO_2$ concentration.



## 4 Discussion and conclusions

It is not surprising that soil $CO_2$ concentration decreases in the deep layers over carbonate bedrock areas, especially at the bottom of soil profiles, as has been observed by many experiments(Buyannovsky and Wagner, 1983; Li et al. 1995; Xu and He, 1996; Liang et al. 2003), and was now supported by this paper. The explanation by some studies (Li et al., 1995) that decreased $CO_2$ is caused by decreased microbe or root respiration in deep soil, is challenged by our data. At first, one important reason leading to the earlier conclusion lies perhaps in the lack of comparative analyses of soil $CO_2$ levels in carbonate and non-carbonate areas. The underlying foundation of soluble carbonate in carbonate areas was not taken into consideration, and, most important, there was no proof or data to support this idea. Secondly, there is no decrease of $CO_2$ in soil profiles of non-carbonate areas (mudstone, basalt, shale or siltstone areas), also it seems to be reasonable to expect $CO_2$ decrease by lower microbe or root respiration rates in deep soil layers of both carbonate or non-carbonate areas. Thirdly, decrease of soil $CO_2$ takes place in 20-30 cm soil layers, and even from the soil surface in some profiles, so it may be unreasonable to attribute $CO_2$ decrease to microbe respiration in such shallow occurrences.

Additionally, soil profile $CO_2$ only decreases in carbonate areas, and SOC content is positively correlated with soil $CO_2$ concentration in non-carbonate areas ($R^2$=0.67-0.85), although there is no significant correlation at some profiles because soil $CO_2$ is not only related with organic carbon, but also with other factors, such as root respiration. Soil $CO_2$ and organic carbon in different depths of carbonate areas are positively correlated with low correlation coefficients, but not in soil profiles of these carbonate areas. This means that organic carbon cannot be responsible for the decreased $CO_2$ concentrations. Furthermore, $CO_2$ consumed by carbonate corrosion leads to uncorrelated re-lationship between soil $CO_2$ and organic carbon levels in carbonate areas. Soil profile pH in carbonate areas always suddenly and sharply decreases at the depth of $CO_2$ decrease, and this can be explained well by carbonate corrosion. Analysis of $\delta^{13}C$ isotope, which mixes into the $CO_2$ in deep soil layers of carbonate bedrock areas (dolomite or limestone) also demonstrates that there is another carbon source, whose $\delta^{13}C$ level is more than -14‰. In soil of clay-stone areas, however, soil $CO_2$ and soil organic carbon have the same $\delta^{13}C$ value. This provides strong evidence that carbonate corrosion occurs, and thus deep soil $CO_2$ is consumed in carbonate areas. Simply stated, our work strongly indicates that carbonate corrosion leads to the decrease of soil profile $CO_2$ concentration in areas with carbonate bedrock.

Further, a mathematical model of soil $CO_2$ transfer was developed, showing that soil $CO_2$ concentration can be roughly expressed as a linear or parabolic increase with soil depth. The linear or parabolic increase can be demonstrated, strongly supported by both field data and the models. Soil $CO_2$ concentration data, collected in non-carbonate areas or in the surface soil of carbonate areas, provide additional confirmation. In the deep soil of carbonate areas, however, especially at the rock-soil interface, the simulated values are always higher than the field measurements. All of these points may also indicate that carbonate corrosion occurs in the deep soil, and that apart of soil $CO_2$ is consumed by carbonate corrosion. In addition, the decreased $CO_2$ concentration caused by carbonate corrosion can be evaluated by the subtraction of measured and simulated $CO_2$. The decreased $CO_2$ concentration is related closely to deep soil pH and CaO content of carbonate rock (correlation coefficients, respectively, $R^2$=0.97 and 0.41), together with farming activities, but not with deep soil water content and MgO content of carbonate. These results and conclusions can be supported by experiments, and are widely accepted by karst scholars, who add validity to our results and conclusions.

The carbon cycle in karst areas has attracted big attention because of the imbalance of the



global carbon cycle, and in recent years there has been a search to resolve the missing sink
related to the absorbing and releasing of carbon in $CaCO_3$ systems (Jiang and Yuan, 1999).
Experiments and calculations indicate that $1.774 \times 10^7$ t of carbon are absorbed by karstification
in China, and that $2.2 \times 10^8 \sim 6.08 \times 10^8$ t of carbon are drawn back from the atmosphere
worldwide every year (Jiang and Yuan, 1999). It is obviously significant with regard to the
increasing atmospheric temperature. Soil, as an important carbon storage area, is of great
importance to atmospheric $CO_2$ concentration, and slight variations may impose great effects
on global carbon cycle. Several factors affecting soil $CO_2$ concentration have been discussed,
such as environmental ones (soil temperature, moisture, water content, etc.), microbe activities,
and human activities, but no published details about the effect of carbonate corrosion on soil $CO_2$
concentration can be found. Our study argues that deep soil $CO_2$ concentrate in carbonate areas
is obviously decreased, especially at the rock-soil interface, and that this is mainly caused by
carbonate corrosion. If this conclusion is correct, then naturally the atmospheric $CO_2$ levels in
carbonate areas should be affected by the corrosion, and this should be very significant in the
hunting for the "missing sink".
**Author contributions.** CQ developed the work and wrote the paper.
**Competing interests**. The author declare that he has no conflict of interest
**Acknowledgments.** The author express the heartfelt thanks to the staff of the Zhaotong Bureau of Sciences
and Technology, and the IGSNRR in gathering the samples, as well as Professor C B. Wood, Department of
Biology Faculty, Providence College, USA for correcting English grammar. This work was financially
supported by the Chinese National Key Natural Science Foundation (Grant No. 90202017), the Natural Science
Fund of Shandong Province (ZR2018MD012), 2017 Special Fund forI Scientific Research of Shandong
Coalfield Geologic Bureau [2017(10)].

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
