# Peer review of "Characteristics of soil profile CO2 concentrations in karst areas and its significance for global carbon cycles and climate change"

_Earth System Dynamics, 2018_

## Short Comment (SC1) · 13 Feb 2019

There are several citations where the authors first and last names have been reversed such as the paper titled Origin and distribution of carbon dioxide in the unsaturated zone of the southern High Plains of Texas should be W.W. Wood and M.J Petraitis and not Warren et al.. This is also the case for Jeremy et al. and it looks like there are others.

Also, the point that Owens et al. are making is that there is a missing sink of carbon which is based on mapping and carbon isotopes. As they state this cannot be due to carbonate burial as the fractionation from seawater is very minor. Additionally, weathering of carbonated would have limited effects on the isotope system as well. Thus, I would suggest using the citation of Owens et al. (currently Jeremy et al.) for missing carbon sink should be discussed in a more thorough manner.
* * *

---

## Referee Comment (RC1) · Anonymous Referee #1 · 17 Mar 2019

Manuscript No: esd-2018-82 Journal: Earth System Dynamics

Title: "Characteristics of soil profile CO2 concentrations in karst areas and its significance for global carbon cycles and climate change" by Quio Chen

The author of this manuscript presented analysis on the observed soil profiles of CO2 concentrations over Zhaotong City, Yunnan Province. The manuscript provides with the necessary information on understanding the relationship between soil CO2 concentrations with soil depth. This analysis further discussed on the correlations between soil organic carbon and CO2 soil concentrations, which is negatively correlated. Overall manuscript could makes a significant contribution to the scientific research by providing vital information on soil CO2 concentrations in carbonate and non-carbonate areas. However, I believe that the original manuscript needs some analysis and changes to improve its quality. Hence, I will go with decision as "Accept with MINOR revision" and corrections should be strictly implemented before it is accepted for publication.

General Comments Page No.1, Line No.39-41: What is the importance of this Karst areas for measuring soil CO2 concentrations, explain it with the proper references Page No.1, Line No.41-43: Re-write the sentence "Therefore, some workers have looked for the "missing sink" within the absorbed and released carbon in karst systems, and the estimated values reach a dominating part (almost 1/3) of the missing sink (Jiang and Yuan, 1999)". What does the mean of " some workers" in the following sentence, which needs to be correct. Page No.2, Line No. 52-54: The science questions which posed by the author needs be to addressed properly. Page No. 2, Line No. 87-89: Re-write the sentence, It's not clear Page No.6, Line No. 176-181: Discussions are not properly addressed. Re-write the sentence Specific Comments: 1. I suggest author of this manuscript to look into and revise the abstract in accordance with the results of this manuscript, which is missed in the current version. 2. I suggest to author to improve the quality of the figures like increase the font sizes on both axis, label sizes which needs to redraw for a the better representation and visualization 3. Page 14, section 3.7: I suggest the author to look into it and can be revised the sub-section as " The major controlling factors of decreased CO2 concentrations"

---

## Referee Comment (RC2) · Anonymous Referee #2 · 23 May 2019

I have carefully read the paper titled "Characteristics of soil profile CO2 concentrations in karst areas and its significance for global carbon cycles and climate change" by Qiao Chen. The author measured soil profile CO2 concentrations, and compared their characteristics in areas with carbonate bedrocks and non-carbonate bedrocks. The observed phenomenon that soil CO2 was consumed by carbonate corrosion during karstification in this paper is very interesting and scientific. The evaluation of the decreased CO2 concentration by carbonate corrosion seems to be logical and reasonable. All these ideas presented provide significance information for the global carbon cycle and is interesting enough although there are still the following specific comments:

1. Page 2 Line 51, "no detail documents detailing. . .. . ..", "detail" is repeated here; Page 2 Line 81-82, "among which only Ordovician and Permian Strata appear together", the sentence is not clear, what does this sentence means?; Page 4 Line 129, what does "significantly congressed at 0.005 level, it should be " significant correlation at 0.005 level" or not?; Page 6 Line 169-170, " Results are given. . .. . ...areas", the sentence is not clear, rewrote the sentences. Page 6 Line 177, "which means" is repeated here; 2. I would like to suggest the author show the suitable map for Figure. 1. Maybe the exact location can be shown by the general location and then in 2 or 3 step so as to be acquainted by other researchers. 3. The unit for the ordinate in Figure.6 should be provided.

Overall, the paper is well written and in good shape. It is suitable to be published after the minor revision.

---

## Author Comment (AC1) · 1 Jul 2019

I would like to show thanks to Jeremy Owens for his comments. I have corrected as following: Question: There are several citations where the authors first and last names have been reversed. Response: Indeed, the author first and last names of several references were reversed. Now, I have carefully checked every reference and revised them. Question: additionally, weathering of carbonated would have limited effects on the isotope system as well. Thus, I would suggest using the citation of Owens et al. (currently Jeremy et al.) for missing carbon sink should be discussed in a more thorough manner. Response: As far as the missing carbon sink, I checked some new

information. The referece (Sundquist, 1993) is suitable. The carbon isotope system may be related to complex conditions. For example, Hoefs J (1997) presented that the $CO_2$ from carbonate corrosion has carbon isotope of -3‰+1‰ under acid conditions, which is significantly higher than that of SOC. The studied area is reported to be located an area with acid rain.
* * *

---

## Author Comment (AC2) · 1 Jul 2019

Thanks indeed for the reviewer' useful comments, which help us to improve the Ms. And thanks the reviewer to recommend to accept with minor revision. I have revised the manuscript according to the comments: Question:Page No.1, Line No.39-41: What is the importance of this Karst areas for measuring soil CO2 concentrations, explain it with the proper references. Response: Yes, there should be some proper references (Li and Yuan, 1995; Martin et al. 2013) Question: Page No.1, Line No.41-43: Re-write the sentence "Therefore, some workers have looked for the "missing sink" within the absorbed and released carbon in karst systems, and the estimated values reach a

dominating part (almost 1/3) of the missing sink (Jiang and Yuan, 1999)". What does the mean of " some workers" in the following sentence, which needs to be correct. Response: "some workers" should be " some scholars" Question: Page No.2, Line No. 52-54: The science questions which posed by the author needs be to addressed properly. Response: Yes, we have addressed it again, and some expression was revised. Question: Page No. 2, Line No. 87-89: Re-write the sentence, It's not clear Page Response: The sentence is changed into " In order to comprehensively reveal characteristics of soil $CO_2$ concentration in karst area, soil profiles of different stratigraphic units and vegetation types were selected. And profiles in carbonate or non-carbonate areas were both involved". Question: No.6, Line No. 176-181: Discussions are not properly addressed. Re-write the sentence. Response: we re-wrote the sentence in such way: The reason for non-significance ($P>0.05$) may be that soil $CO_2$ concentration is related not only to SOC, but also to soil respiration and microbe activities. However, there is no such tendency in carbonate areas as that in shale areas (Table 3), and even those of Banqiao O1m farmland profile and Banqiao P1m(q) shrub profile show increasing tendency. Previous studies in carbonate areas as Shilin, Lunan City and Guizhou Plateau also showed no correlation between $CO_2$ concentration and SOC (Liang et al. 2003). Question:I suggest author of this manuscript to look into and revise the abstract in accordance with the results of this manuscript, which is missed in the current version. Response: I have checked the abstract carefully. The percentage of decreases deep soil $CO_2$ is 5.2-66.3%. Question:I suggest to author to improve the quality of the figures like increase the font sizes on both axis, label sizes which needs to redraw for a the better representation and visualization Response: It is good idea, we have increased the font sizes on both axis, label sizes. Question: Page 14, section 3.7: I suggest the author to look into it and can be revised the sub-section as " The major controlling factors of decreased $CO_2$ concentrations". Response: Yes, I have revised the sub-section.

---

## Author Comment (AC3) · 1 Jul 2019

Thanks for the reviewer's comments and recommendation for possible publication, and I have revised the manuscript as followings: Question: Page 2 Line 51, "no detail documents detailing. . .. . .", "detail" is repeated here; Response: Yes, it is repeated, i have corrected it. Question:Page 2 Line 81-82, "among which only Ordovician and Permian Strata appear together", the sentence is not clear, what does this sentence means?; Response: the part "among which only Ordovician and Permian strata appear together" should be deleted here. Question:Page 4 Line 129, what does "signnificantly congressed at 0.005 level, it should be " signnificant correlation at 0.005 level" or not?;

Response: Yes, it should be "significant correlation at 0.005 level. Question:Page 6 Line 169-170, " Results are given. . .. . ..areas", the sentence is not clear, rewrote the sentences. Response: the sentence should be "Results are given in Fig.4, among which, Fig. 4 a-h, indicate profiles in carbonate areas and Fig. 4 i-l indicate those in non-carbonate (shale) areas." Question:Page 6 Line 177, "which means" is repeated here; Response: Yes, it is deleted. Question:I would like to suggest the author show the suitable map for Figure. 1. Maybe the exact location can be shown by the general location and then in 2 or 3 step so as to be acquainted by other researchers. Response: Good suggestion, We revised the picture, which includes "Yunan province" "Zhaotong City" and " the locations of Zhengxiong and Weixing county" Question:The unit for the ordinate in Figure.6 should be provided Response: we have added the unit(‰.